# Three-Dimensional Digital Zooming of Integral Imaging under Photon-Starved Conditions

**DOI:** 10.3390/s23052645

**Published:** 2023-02-28

**Authors:** Gilsu Yeo, Myungjin Cho

**Affiliations:** Research Center for Hyper-Connected Convergence Technology, School of ICT, Robotics and Mechanical Engineering, Institute of Information and Telecommunication Convergence (IITC), Hankyong National University, 327 Chungang-ro, Anseong 17579, Kyonggi-do, Republic of Korea

**Keywords:** digital zooming, N observations photon counting integral imaging, photon-starved conditions, synthetic aperture integral imaging, three-dimensional integral imaging, volumetric computational reconstruction

## Abstract

In this paper, we propose new three-dimensional (3D) visualization of objects at long distance under photon-starved conditions. In conventional three-dimensional image visualization techniques, the visual quality of three-dimensional images may be degraded because object images at long distances may have low resolution. Thus, in our proposed method, we utilize digital zooming, which can crop and interpolate the region of interest from the image to improve the visual quality of three-dimensional images at long distances. Under photon-starved conditions, three-dimensional images at long distances may not be visualized due to the lack of the number of photons. Photon counting integral imaging can be used to solve this problem, but objects at long distance may still have a small number of photons. In our method, a three-dimensional image can be reconstructed, since photon counting integral imaging with digital zooming is used. In addition, to estimate a more accurate three-dimensional image at long distance under photon-starved conditions, in this paper, multiple observation photon counting integral imaging (i.e., N observation photon counting integral imaging) is used. To show the feasibility of our proposed method, we implement the optical experiments and calculate performance metrics, such as peak sidelobe ratio. Therefore, our method can improve the visualization of three-dimensional objects at long distances under photon-starved conditions.

## 1. Introduction

Three-dimensional (3D) visualization of objects at long distances on photon-starved conditions has been a great challenge in many applications, such as military, astronomy, and observing wild animals. In the military case, a defense or reconnaissance that searches enemies at long distances in the day or night is required. In astronomy, observing stars at billions of light years of distance is a critical problem. In addition, observing wild animals, which are nocturnal and have much wariness, is also needed.

However, it is difficult to visualize the three-dimensional objects, which are located at long distances by conventional imaging methods, since lateral and longitudinal resolutions of the image at long distance may be reduced due to the limitation of optical devices and the image sensor. When a camera takes a picture, the object at long distance in the scene has less pixels than a close one. Therefore, lateral and longitudinal resolutions (i.e., three-dimensional resolution) of the image for objects at long distance are reduced. To visualize three-dimensional objects at long distance, integral imaging [1,2,3], which was first proposed by G. Lippmann, can be utilized. It uses two-dimensional (2D) images with different perspectives captured by lenslet array or camera array, where these images are referred to as elemental images. Integral imaging can provide full parallax and continuous viewing points of three-dimensional objects without any viewing glasses and coherent light sources [1,2,3,4,5,6,7,8]. However, due to the limitation of three-dimensional resolution for three-dimensional objects at long distances, the visual quality of three-dimensional images at long distances may be degraded. In addition, this resolution problem may be critical under photon-starved conditions. Because an image sensor detects less photons, which have the information of an object at a long distance under photon-starved conditions, elemental images may not have the information of the object. That is, the visual quality of three-dimensional images may be more degraded under photon-starved conditions.

To visualize three-dimensional objects under photon-starved conditions, photon counting integral imaging [9,10,11] has been proposed. It can make a computational model of a photon detector by statistical distribution, such as Poisson distribution, because photons occur rarely in unit time and space [11]. In addition, for three-dimensional image reconstruction, statistical estimation methods, such as maximum likelihood estimation (MLE) [9,10,11] or Bayesian approaches [12,13,14], are utilized. However, photon counting integral imaging may not estimate the accurate three-dimensional images for objects at long distances under photon-starved conditions, since object images may have low resolution and an insufficient number of photons. Therefore, to solve these problems, a new three-dimensional image visualization technique is required.

In this paper, we propose three-dimensional digital zooming of integral imaging under photon-starved conditions. It can magnify region of interest (ROI) in elemental images captured by synthetic aperture integral imaging (SAII) [15]. Then, three-dimensional images at long distances can be obtained by volumetric computational reconstruction (VCR) [16,17,18,19,20,21,22,23] and photon counting integral imaging [9,10,11,12,13,14]. Under photon-starved conditions, photons can be detected throughout the scene by computational photon counting imaging, which may cause the degradation of resolution for objects due to lack of the number of photons. However, in our method, photon counting imaging is utilized only in ROI of elemental images to visualize three-dimensional images at long distances. Therefore, more photons can be extracted from the ROI of elemental images. In addition, multiple observations of photon counting imaging is considered in our method, where this method is called “N observation photon counting imaging”, which improves the visual quality of the images under photon-starved conditions, since photons are detected randomly for each observation, and multiple observation can increase the number of samples. Additionally, to estimate more accurate three-dimensional images under photon-starved conditions, statistical estimation methods, such as maximum likelihood estimation (MLE), are used in our method.

This paper is organized as follows. We describe the basic concept of optical and digital zooming and integral imaging in Section 2. Then, we introduce the computational photon counting method and our proposed method in Section 3. To show the feasibility of our proposed method, we show the experimental results in Section 4. Finally, we make a conclusion with summary in Section 5.

## 2. Related Work

In this section, we briefly present the concept of optical and digital zooming and integral imaging.

### 2.1. Zooming

In general, to visualize objects at a long distance, two zooming methods, such as optical and digital zooming, can be applied. In optical zooming, objects at long distances can be magnified by modifying lenses’ positions (i.e., use of zoom lens). This can obtain the best visual quality of the image without any interpolation methods. However, it needs more complicated optical devices and is more expensive. On the other hand, in digital zooming, it is easy to visualize objects at long distances by using various interpolation methods. Thus, it is more cost-effective and simpler. However, the visual quality of the image is worse than the image by optical zooming. Recently, two digital zooming methods have been utilized, being interpolation and deep learning based methods. The interpolation method can magnify the image by digital image processing.

In general interpolation methods, there are “nearest”, “bilinear”, and “bicubic” methods. These methods use near pixels to interpolate a pixel value at a new location [24]. The difference between their methods is the number of near pixels. The “nearest” interpolation method uses the single nearest pixel, the “bilinear” interpolation method utilizes the four nearest pixels and distances for interpolation, and the “bicubic” interpolation method interpolates the new pixel by using 16 pixels, which are weighted. The “nearest” method is faster to process than others, but degrading resolution is a disadvantage. The “bicubic” interpolation method has the best result of interpolation, but speed is slow. On the other hand, both the quality and speed of the “bilinear” method are more appropriate than others. Recently, several novel interpolation methods were used, such as the “Lagrange-Chebyshev” and the “de la Vallée-Poussin” [25,26]. In addition, a deep learning-based method, which can estimate the magnified image by a neural network [27], has been reported. This deep learning-based method can estimate more detail of the zooming region with high magnification ratio in real-time. However, a huge amount of data and time are required to train the neural network.

Cameras in state-of-the-art smart phones utilize interpolation with deep learning to have the best quality of images at long distances. In this paper, we use the interpolation method for digital zooming of objects at long distance because it is easy and fast for implementation. To visualize three-dimensional objects at long distances, we introduce integral imaging, which will be described in the next section.

### 2.2. Integral Imaging

Integral imaging is a passive three-dimensional visualization technique. Figure 1 illustrates the basic concept of integral imaging by lenslet array and camera array. In the pickup stage, it can record multiple two-dimensional images with different perspectives (i.e., elemental images) from the scene by lenslet array or camera array. In optical reconstruction or display, a three-dimensional image can be displayed by projecting elemental images through the homogeneous lenslet array used at pickup stage. Additionally, a three-dimensional image can be reconstructed by back-projecting elemental images through virtual pinhole arrays on the reconstruction plane in computational reconstruction, where it is called volumetric computational reconstruction (VCR) [16,17,18,19,20,21,22,23].

In integral imaging, two different pickup methods can be used, such as lenslet array and camera array. Lenslet array-based integral imaging is shown in Figure 1, and it can record the elemental images by single shot, but the resolution of these elemental images is low since the resolution of the image sensor is divided by the number of lenslets. On the other hand, camera array-based integral imaging is shown in Figure 1, which can record the elemental images with the same resolution as the image sensor. This camera array-based integral imaging is called synthetic aperture integral imaging (SAII) [15]. However, it is difficult to align and synchronize the camera array. In addition, since moving a single camera to avoid these problems is slow for pickup, it may impossible for capturing the dynamic scene.

To visualize three-dimensional images computationally, elemental images are back-projected through the virtual pinhole array on the reconstruction plane as shown in Figure 2. Reconstruction is the reverse process of pickup stage. When elemental images are back-projected through the virtual pinhole array on the reconstruction plane, they have different shifting pixels at various reconstruction depths. Shifting pixel value can be found as the following [23]
(1)$$\Delta S_{x}=\frac{N_{x}\times p_{x}\times f}{c_{x}\times z},\;\Delta S_{y}=\frac{N_{y}\times p_{y}\times f}{c_{y}\times z}$$
where $\Delta S_{x}$, $\Delta S_{y}$ are the number of shifting pixels for elemental images in *x* and *y* directions, $c_{x}$, $c_{y}$ are the sensor size, and $N_{x},N_{y}$ are the number of pixels for each elemental image, respectively. $p_{x}$, $p_{y}$ are the pitch between elemental images, *f* is the focal length of the virtual pinhole, and *z* is the reconstruction depth. Using shifting pixel values calculated by Equation (Equation 1), elemental images are overlapped with each other on the reconstruction plane. Then, three-dimensional images are obtained by [23]
(2)$$\Delta S_{x}^{k}=\left\lfloor k\times \Delta S_{x}\right\rceil ,\;\mathrm{for}\;k=0,1,2,3,\cdots ,K-1$$
(3)$$\Delta S_{y}^{l}=\left\lfloor l\times \Delta S_{y}\right\rceil ,\;\mathrm{for}\;l=0,1,2,3,\cdots ,L-1$$
(4)$$O\left(x,y,z\right)=\sum_{k=0}^{K-1}\sum_{l=0}^{L-1}𝟙\left(x+\Delta S_{x}^{k},\;y+\Delta S_{y}^{l}\right)$$
(5)$$I\left(x,y,z\right)=\frac{1}{O(x,y,z)}\sum_{k=0}^{K-1}\sum_{l=0}^{L-1}E_{kl}\left(x+\Delta S_{x}^{k},\;y+\Delta S_{y}^{l}\right)$$
where K,L are the number of elemental images in *x* and *y* directions, $\Delta S_{x}^{k},\Delta S_{y}^{l}$ are rounded shifting pixels of *k*th column and *l*th row elemental image, O(x,y,z) is the overlapping matrix for the reconstructed three-dimensional image at the reconstruction depth *z*, 𝟙 is the ones matrix, $E_{kl}$ is *k*th column and *l*th row elemental image, and I(x,y,z) is the reconstructed three-dimensional image. Equations (1)–(5) are utilized for VCR [23].

$N_{x}$, $N_{y}$, $p_{x}$, $p_{y}$, *f*, $c_{x}$, and $c_{y}$ are constant values, and only *z* is variable. Thus, when the reconstruction depth increases, the value of shifting pixel and depth resolution are changed. Our method uses a digital zooming method to enhance the shifting pixel and depth resolution in computational reconstruction of integral imaging.

However, under photon-starved conditions, the detail information of ROI of objects at long distance may be lost due to lack of the number of photons. To enhance the visual quality of three-dimensional images at long distances, we utilize computational photon counting integral imaging, as described in next subsection.

## 3. Three-Dimensional Visualization of Objects at Long Distances under Photon-Starved Conditions

### 3.1. Computational Photon Counting Imaging

Under photon-starved conditions, it is difficult to record the information of objects from the scene using conventional imaging methods due to lack of the number of photons. To overcome this problem, computational photon counting imaging is utilized in our method. Photons can be detected by Poisson random process under these conditions because photons may occur rarely in unit time and space [11]. Computational photon counting imaging can be described as follows [9,10,11,12,13,14]
(6)$$\lambda \left(x,y\right)=\frac{N_{p}I\left(x,y\right)}{\sum_{x=1}^{N_{x}}\sum_{y=1}^{N_{y}}I\left(x,y\right)}$$
(7)C(x,y)|λ(x,y)∼Poisson(λ(x,y))
where λ is the normalized irradiance of the elemental image, which has unit energy, I(x,y) is the recorded two-dimensional image, C(x,y) is the photon counting image, and $N_{p}$ is the expected number of photons from the elemental image. Since λ has unit energy by Equation (Equation 6), $N_{p}$ photons can be extracted randomly from the recorded two-dimensional image by Equation (Equation 7). To obtain a three-dimensional image under photon-starved conditions, photon counting integral imaging [11] can be used. At first, to estimate the recorded two-dimensional image from photon counting image, maximum likelihood estimation (MLE) can be applied as follows [9,10,11]
(8)$$L\left(\lambda_{kl}\right)=\prod_{k=0}^{K-1}\prod_{l=0}^{L-1}\frac{\lambda_{kl}{\left(x,y\right)}^{C_{kl}}e^{-\lambda_{kl}\left(x,y\right)}}{C_{kl}!}$$
(9)$$l\left(\lambda_{kl}\right)∼\sum_{k=0}^{K-1}\sum_{l=0}^{L-1}C_{kl}\ln\lambda_{kl}\left(x,y\right)-\sum_{k=0}^{K-1}\sum_{l=0}^{L-1}\lambda_{kl}\left(x,y\right)$$
(10)$$\frac{\partial l(\lambda_{kl})}{\partial \lambda_{kl}}=\frac{C_{kl}}{\lambda_{kl}\left(x,y\right)}-1=0$$
(11)$${\hat{\lambda }}_{kl}=C_{kl}$$
where $\lambda_{kl}$ is the normalized irradiance of *k*th column *l*th row elemental image, L(),l() are the likelihood and log-likelihood functions, and $\hat{\lambda_{kl}}$ is the *k*th column *l*th row-estimated two-dimensional image for the scene, respectively. By using MLE and VCR, a three-dimensional image can be visualized under photon-starved conditions, as follows [9,10,11]
(12)$$I_{p}\left(x,y,z\right)=\frac{1}{O(x,y,z)}\sum_{k=0}^{K-1}\sum_{l=0}^{L-1}{\hat{\lambda }}_{kl}\left(x+\Delta S_{x}^{k},y+\Delta S_{y}^{l}\right)$$
where $I_{p}\left(x,y,z\right)$ is a three-dimensional image under photon-starved conditions obtained by photon counting integral imaging. However, three-dimensional objects at long distance under photon-starved conditions may not be visualized due to lack of the number of photons in ROI of the scene. Therefore, in this paper, we present digital zooming and photon counting integral imaging for three-dimensional objects at long distances under photon-starved conditions, as described in the next section. In our method, we use different VCRs for digital zooming, where shifting pixels are recalculated because elemental images are cropped by ROI of the scene. In addition, to improve the visual quality of photon counting image, N observation photon counting imaging is proposed.

### 3.2. Three-Dimensional Digital Zooming of Integral Imaging and N Observation Photon Counting Integral Imaging under Photon-Starved Conditions

As mentioned earlier, the conventional integral imaging has the problems that resolution of three-dimensional image and depth resolution of object at long distance are worst. In addition, resolution of three-dimensional images for objects at long distances under photon-starved conditions is also much worse. To solve these resolution problems, in our method, we utilize digital zooming and VCR to visualize three-dimensional images at long distances, as illustrated in Figure 3.

As mentioned earlier, digital zooming is applied to obtain new elemental images by cropping ROI from the original elemental images before overlapping the elemental images with shifting pixels on the reconstruction plane. When ROI is cropped from the original elemental image, the aspect ratio of the image is preserved as the original elemental images. Then, new elemental images are interpolated, and their size is the same as the original elemental images. Therefore, new VCR for three-dimensional objects at long distances, considering digital zooming, can be written as follows
(13)$$N_{x}^{\prime }=\frac{1}{m}\times N_{x},\;N_{y}^{\prime }=\frac{1}{m}\times N_{y},\;\left(m>1\right)$$
(14)$$N_{x}^{\prime }:N_{x}=z^{\prime }:z\to z^{\prime }=\frac{1}{m}\times z$$
(15)$$\Delta S_{x}=\frac{N_{x}\times p_{x}\times f}{c_{x}\times z}\to \Delta S_{x}^{\prime }=\frac{N_{x}\times p_{x}\times f}{c_{x}\times z^{\prime }}$$
(16)$$\Delta S_{y}=\frac{N_{y}\times p_{y}\times f}{c_{y}\times z}\to \Delta S_{y}^{\prime }=\frac{N_{y}\times p_{y}\times f}{c_{y}\times z^{\prime }}$$
(17)$$\Delta S_{x}^{k^{\prime }}=\left\lfloor k\times \Delta S_{x}^{\prime }\right\rceil ,\;\mathrm{for}\;k=0,1,2,\cdots ,K-1$$
(18)$$\Delta S_{y}^{l^{\prime }}=\left\lfloor l\times \Delta S_{y}^{\prime }\right\rceil ,\;\mathrm{for}\;l=0,1,2,\cdots ,L-1$$
(19)$$\tilde{O}\left(x,y,z\right)=\sum_{k=0}^{K-1}\sum_{l=0}^{L-1}𝟙\left(x+\Delta S_{x}^{k^{\prime }},y+\Delta S_{y}^{l^{\prime }}\right)$$
(20)$$\tilde{I}\left(x,y,z\right)=\frac{1}{\tilde{O}\left(x,y,z\right)}\sum_{k=0}^{K-1}\sum_{l=0}^{L-1}{\tilde{E}}_{kl}\left(x+\Delta S_{x}^{k^{\prime }},y+\Delta S_{y}^{l^{\prime }}\right)$$
where $N_{x}^{\prime },N_{y}^{\prime }$ are width and height of the cropped ROI from the original elemental image, *m* is real value of zooming ratio, which is bigger than one, $z^{\prime }$ is a zooming distance by digital zooming, $\Delta S_{x}^{\prime },\Delta S_{y}^{\prime }$ are the number of shifting pixels for new elemental image by digital zooming, $\Delta S_{x}^{k^{\prime }},\Delta S_{y}^{l^{\prime }}$ are the rounded number of shifting pixels of *k*th column *l*th row new elemental image by digital zooming, $\tilde{O}\left(x,y,z\right)$ is the new overlapping matrix by digital zooming, ${\tilde{E}}_{kl}$ is the *k*th column and the *l*th row of new elemental image by digital zooming, and $\tilde{I}\left(x,y,z\right)$ is the three-dimensional image by digital zooming, respectively. The size of ROI is always less than the original elemental images, and new elemental images are interpolated with the same size as the original elemental images. Thus, through Equations (13)–(20), the shifting pixels are changed, and a three-dimensional image by digital zooming can be obtained.

Three-dimensional objects at long distances under photon-starved conditions can be visualized by combining our digital zooming and computational photon counting integral imaging. However, since new elemental images have limited resolution, photon counting integral imaging may not produce three-dimensional images with sufficient visual quality. Therefore, in this paper, we propose N observation photon counting imaging as depicted in Figure 4. In this method, photon counting images are generated and estimated N times. Then, they are accumulated and averaged in relation to each other. Finally, the estimated images with better visual quality can be obtained, since the number of samples for each photon counting image increases. To verify the feasibility of our method, we describe our experimental setup and results in the next section.

## 4. Simulation and Experimental Results

In this section, we present our simulation and experimental setup for obtaining the elemental images by SAII. Then, we show the simulation and experimental results to prove the feasibility of our method.

### 4.1. Simulation and Experimental Setup

Before we implement the optical experiment, we implemented computer simulation by “Blender”. We set the simulation environment as depicted in Figure 5.

The object is ‘ISO-12233’, which is located at 1600 mm from camera array. In this simulation setup, focal length and sensor size of virtual camera are 50 mm and 36 mm (H) × 24 mm (V), respectively. The size of elemental image is 1080 (H) × 720 (V), and pitch between cameras is 2 mm. Total number of elemental images is 5 (H) × 5 (V). Through this setup, we evaluate the performance of interpolation methods, such as “nearest”, “bilinear”, and “bicubic”.

To record the elemental images by SAII, experimental setup is illustrated in Figure 6. The yellow helicopter and the fire truck are used as three-dimensional objects at close distance (40 mm) and long distance (520 mm), respectively, because we require two different objects, which are located at close and long distances to prove our zooming ability. In this setup, Nikon D850 is used as the image sensor, and Nikon DX AF-S Nikkor 18–55 mm is used as the camera lens. The sensor size is 36 mm (H)× 24 mm (V), and the focal length of the camera lens is set to 18 mm. Total number of the recorded elemental images is 5 (H)× 5 (V), each elemental image has 5408 (H)× 3600 (V) pixels, and the pitch between cameras is 0.5 mm. Then, for digital zooming, the size of ROI is set to 676 (H) × 450 (V) pixels, which is eight times smaller than the original elemental image. That is, the magnification ratio is eight. Thus, the distance between objects and camera is reduced as magnification ratio, which is the ratio between size of the elemental image and cropped ROI when a three-dimensional image is reconstructed by VCR with digital zooming. Figure 7a illustrates the center image among elemental images captured by our experimental setup, and Figure 7b is the new elemental image for digital zooming, which is the cropped and interpolated ROI of Figure 7a.

To show the availability of visualization under photon-starved conditions, we captured the elemental images under these conditions as shown in Figure 8. The elemental image under photon-starved conditions is shown in Figure 8a, which was captured by our experimental setup, where the exposure time of the camera was 13 s to obtain the photons, including the object information at long distance. Then, for digital zooming, new elemental image was generated as shown in Figure 8d. To estimate the elemental images under these conditions, photon counting imaging with maximum likelihood estimation was used, where the numbers of extracted photons are 97,344 and 973,440. Finally, photon counting images without digital zooming and with digital zooming were obtained, as shown in Figure 8b,c and Figure 8e,f, respectively.

### 4.2. Experimental Result

Figure 9 shows three-dimensional images of simulation results. Figure 9a–c are the reconstructed three-dimensional images by digital zooming VCR with “bicubic”, “bilinear”, and “nearest” interpolation methods. Their depths are the same as 200 mm, and they were digitally zoomed with magnification ratio eight.

To verify our method, we calculate the peak sidelobe ratio (PSR) of correlation via different depths as the performance metric. PSR of the correlation peak is defined as the number of standard deviation by which the peak exceeds the mean value of the correlation surface. It can be calculated by [28]
(21)$$PSR=\frac{max\left[c\left(x\right)\right]-\mu_{c}}{\sigma_{c}}$$
where $\mu_{c}$ is the mean of the correlation, and $\sigma_{c}$ is the standard deviation of the correlation. The higher the PSR value is, the better the recognition performance obtained.

To calculate the PSR value, the reference image, which is reconstructed at 200 mm, is used. Magnification is eight, 1600÷8=200 by using Equation (Equation 14). Table 1 shows the CPU time for obtaining three-dimensional images at 200mm depth and PSR value. The specification of the computer used in this simulation is AMD Ryzen 7 1700X Eight-Core Processor 3.40 GHz, 16GB of RAM. CPU time of each interpolation method is almost the same. “Nearest” method is the fastest, but its PSR value at focusing depth is the lowest. As shown Figure 10, the “nearest” method has the worst PSR value, and the “bicubic” method has the best result. The PSR graphs of “bicubic” and “bilinear” methods via reconstruction depths are very similar to each other. Only “nearest” method does not have the peak PSR value at focusing depth (200 mm). In Table 1, the speed of the “bilinear” method is almost the same as the “nearest” one, but the PSR value is almost the same as the “bicubic” method. In addition, the “bicubic” method is the slowest, and, instead, its PSR value is higher than any other interpolation methods. The “bilinear” method is faster than the “bicubic” method, but it is slower than the “nearest” method. Since it uses four nearest pixels and distances to interpolate at new location, the PSR value of the “bilinear” method is better than the “nearest” one and worse than the “bicubic” method, which uses weighted 16 nearest pixels. Through Figure 9, Table 1, and Figure 10, the “nearest” interpolation method has the worst result among the three interpolation methods, and the “bicubic” method is the best among them. However, the “bilinear” method is as fast as the “nearest” method for interpolation, and this result is almost similar to the “bicubic” method.

Table 2 shows the PSR values of reconstructed three-dimensional images via reconstruction depths by using digital zooming with “nearest”, “bilinear”, and “bicubic” interpolation methods, respectively, where magnification ratio is 8.8. Thus, the reconstruction depth of focused object is 181.82 mm via Equation (Equation 14). However, in Table 2, the peak PSR values (727.47, 720.57 and 705.19) of interpolation methods are located at 186.36 mm, 196.36 mm and 186.36 mm (“bicubic”, “bilinear” and “nearest”), and they are not located at focusing depth. Through Table 2, the location of peak PSR value for each interpolation method has the error.

Figure 11 shows three-dimensional images of our experimental result under conventional conditions. Figure 11a is the reconstructed three-dimensional image by VCR without digital zooming. Here, a fire truck may not be recognized because of low resolution. Figure 11b is the reconstructed three-dimensional image by VCR with digital zooming, where it is noticed that a fire truck can be recognized. To show that our method is better than the conventional method, we cropped “911” characters at various reconstruction depths, respectively. Figure 11c,d were reconstructed at 344 mm and 392 mm by conventional method without digital zooming and Figure 11e,f were reconstructed at 43 mm and 49 mm by our method, respectively. It is noticed that the experimental results by our method have better visual quality than ones by the conventional method.

In Table 3, the PSR values of “bilinear” methods are better than the other two interpolation methods, such as “nearest” and “bicubic” entirely. In addition, this peak value (60.997) is located at 43 mm, which is object depth. In contrast, the other two methods have peak PSR values at wrong object depths. Moreover, since the pixel value by “bicubic” method may be negative, the “bilinear” method has the best result. Through this result, we utilize the “bilinear” interpolation method in our proposed method under photons-starved conditions.

Figure 12 shows the reconstructed three-dimensional images under photon-starved conditions. A fire truck may not be visualized by the conventional method due to low resolution and lack of the number of photons, as shown in Figure 12a–c. In contrast, the reconstructed three-dimensional image by photon counting integral imaging with digital zooming is shown in Figure 12d–f has better visual quality. However, its visual quality is not sufficient for recognition. Therefore, in this paper, we use N observation photon counting integral imaging to improve the visual quality of the reconstructed three-dimensional image under photon-starved conditions. As shown in Figure 12g–i, it is noticed that our method can visualize the reconstructed three-dimensional image at long distance under photon-starved conditions and improve its visual quality for recognition.

In the PSR graph, as shown in Figure 13, the reference image is used as the reconstructed three-dimensional image at 43 mm. The distance 43 mm is calculated by using Equation (Equation 14) because the object is focused at 344 mm, and *m* is 8. The red line in Figure 13 is the PSR result of the conventional method via different depths. Magenta and blue lines in Figure 13 are PSR results of our methods with and without N observation photon counting integral imaging, respectively. In Figure 13, the reconstruction depths by our method are eight times smaller than the ones by conventional method because of digital zooming. Thus, they are modified by multiplying 8 to the reconstruction depths by our method. Table 4 shows the PSR values of the conventional method, digital zooming, and our proposed method, where the conventional method uses integral imaging and computational photon counting imaging only. The tendency of PSR results by the conventional method (red line) have flat and wrong peaks at the reconstruction depth of object. In contrast, PSR results of our method (magenta line) have a sharp peak at the correct reconstruction depth of object. Peak PSR value of conventional method is 32.1834 at 356 mm and 360 mm. On the other hand, peak PSR values of our method with and without N observation photon counting integral imaging are 60.9974 and 41.7063 at 43 mm (i.e., 43 × 8 = 344 mm), respectively. Therefore, through Figure 13 and Table 4, it is noticed that our proposed method can provide a more accurate position of three-dimensional objects at long distance with improved visual quality under photon-starved conditions.

## 5. Conclusions

In this paper, we have presented three-dimensional visualization of objects at long distances under photon-starved conditions using digital zooming and N observation photon counting integral imaging. The conventional method has less lateral and longitudinal resolutions for objects at long distances. In contrast, our method can solve this resolution problem by using digital zooming. In addition, under photon-starved conditions, the three-dimensional information of the object at long distances may be more accurately obtained via digital zooming VCR with N observation photon counting integral imaging. As shown in experimental results, our method had better visual quality of three-dimensional objects at long distances under photon-starved conditions than the conventional method. Therefore, we believe that our method may be used for many applications, such as military, astronomy, and so on. However, our method has a drawback. Because digital zooming utilizes interpolation algorithms for improving the visual quality, it is difficult to zoom the object at very long distance or detail information of object. That is, when the object is located at too long a distance and required magnification is too large, the interpolation method does not work because the nearest pixels may not be helpful for interpolation. Thus, in this case, we require additional optical zooming methods. To solve these problems, we will continue to examine them in future work.

## Figures and Tables

**Figure 1 sensors-23-02645-f001:**
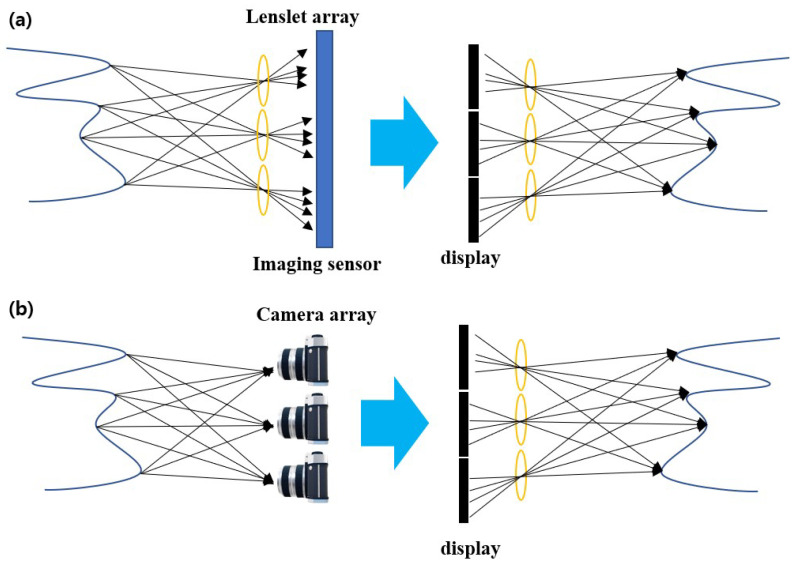
Basic concept of integral imaging by using (**a**) lenslet array and (**b**) camera array.

**Figure 2 sensors-23-02645-f002:**
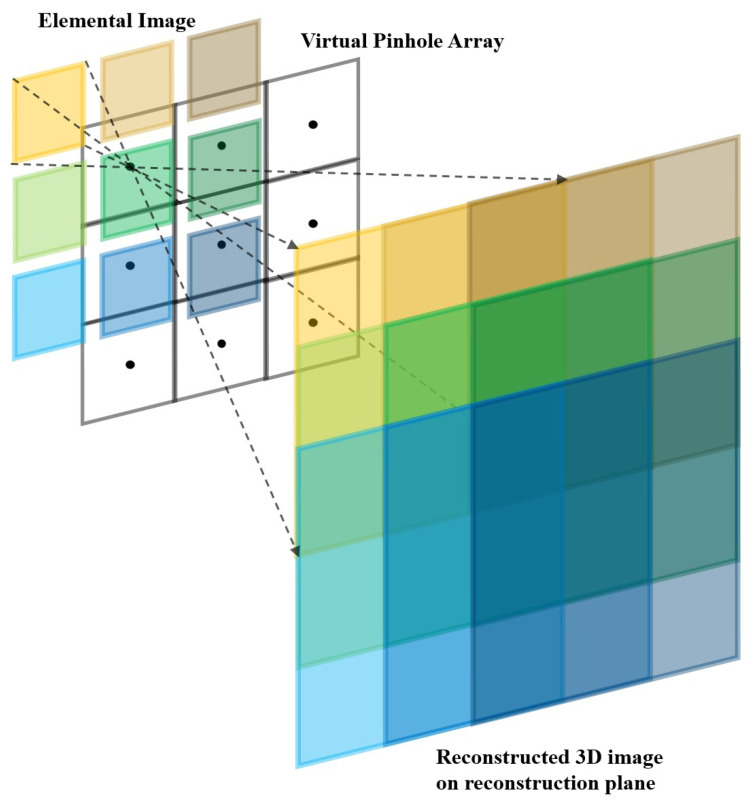
Volumetric computational reconstruction (VCR).

**Figure 3 sensors-23-02645-f003:**
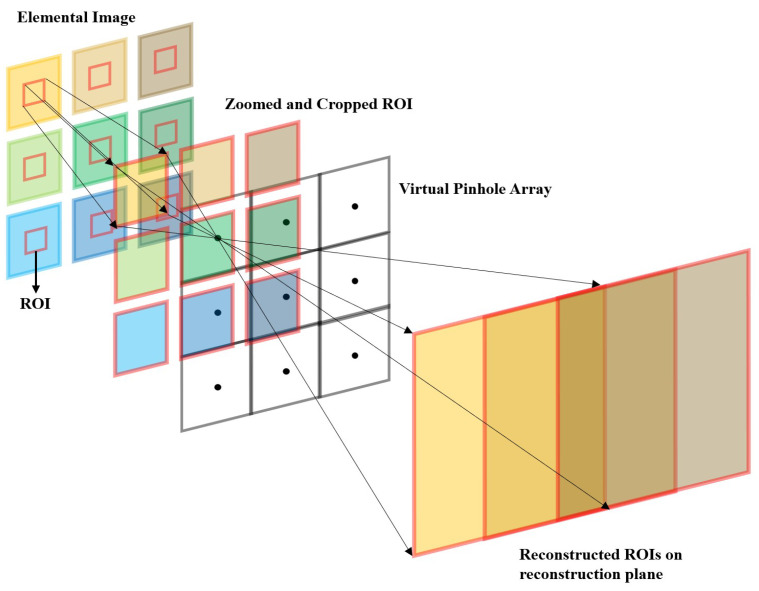
Procedure of our method.

**Figure 4 sensors-23-02645-f004:**
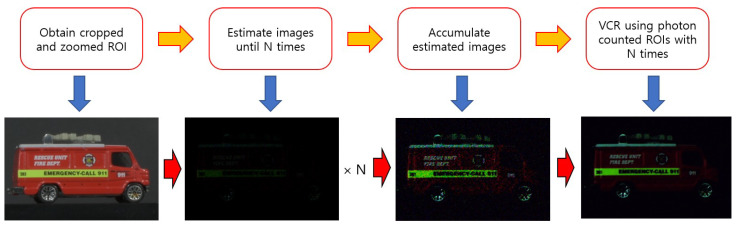
N observation photon counting imaging.

**Figure 5 sensors-23-02645-f005:**
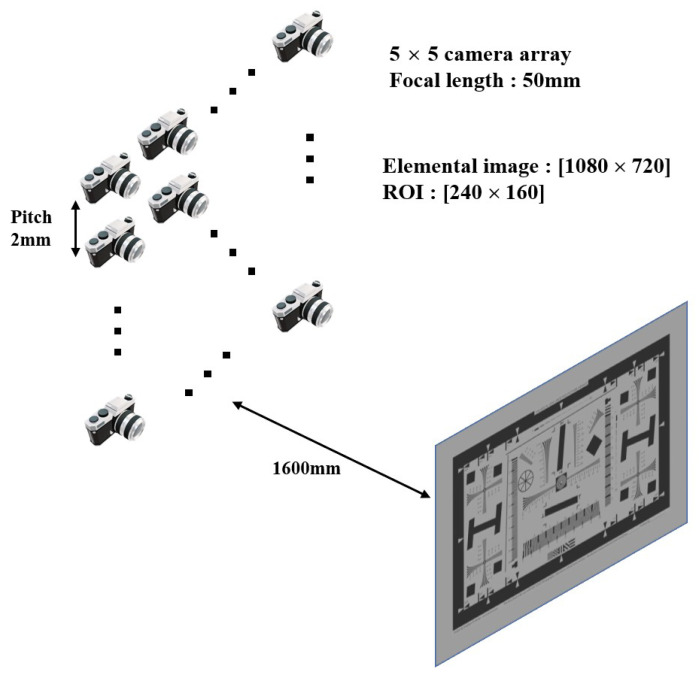
Simulation setup.

**Figure 6 sensors-23-02645-f006:**
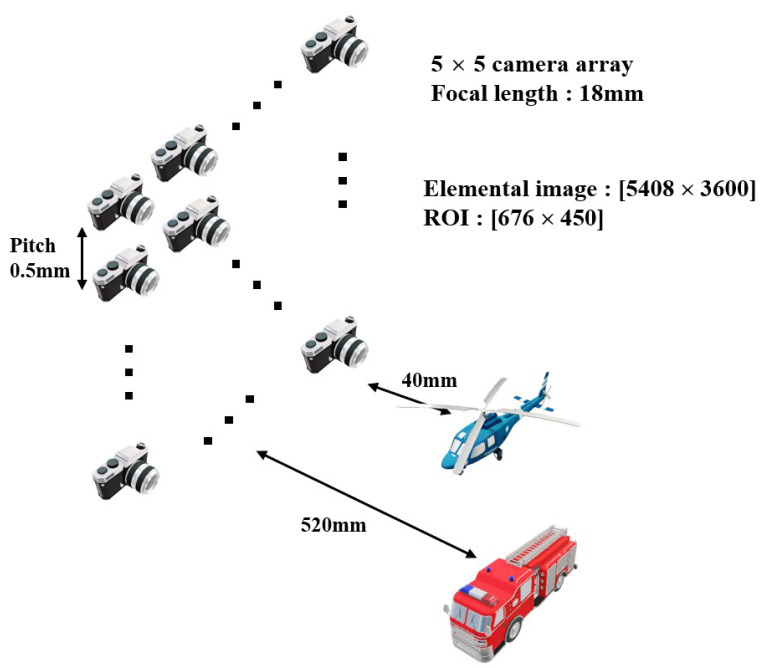
Experiment setup.

**Figure 7 sensors-23-02645-f007:**
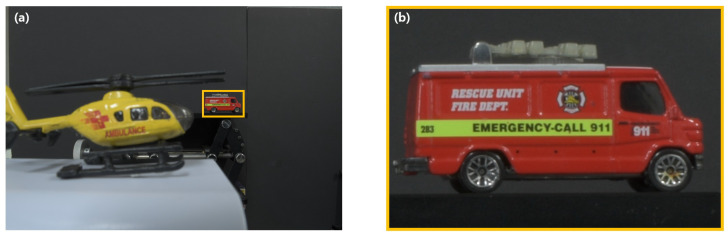
(**a**) 13th elemental image and (**b**) new elemental image for ROI of (**a**).

**Figure 8 sensors-23-02645-f008:**
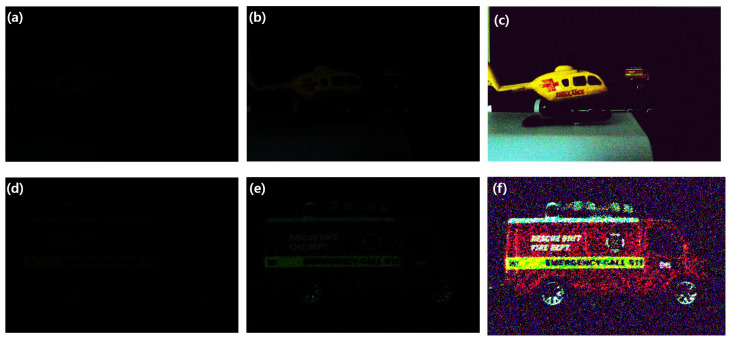
Elemental images and estimated images under photon-starved conditions. (**a**) 13th elemental image, (**b**,**c**) estimated images of (**a**) by computational photon counting imaging with 97,344 and 973,440 photons, respectively, (**d**) new elemental image for ROI of (**a**,**e**,**f**) estimated image of (**d**) by computational photon counting imaging with 97,344 and 973,440 photons, respectively.

**Figure 9 sensors-23-02645-f009:**
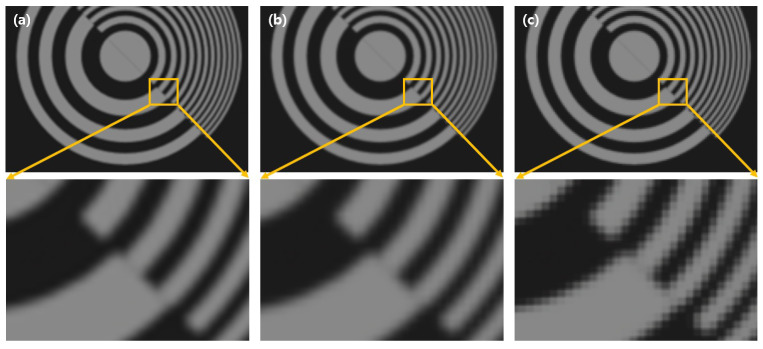
Simulation results with (**a**) bicubic, (**b**) bilinear, and (**c**) nearest, respectively.

**Figure 10 sensors-23-02645-f010:**
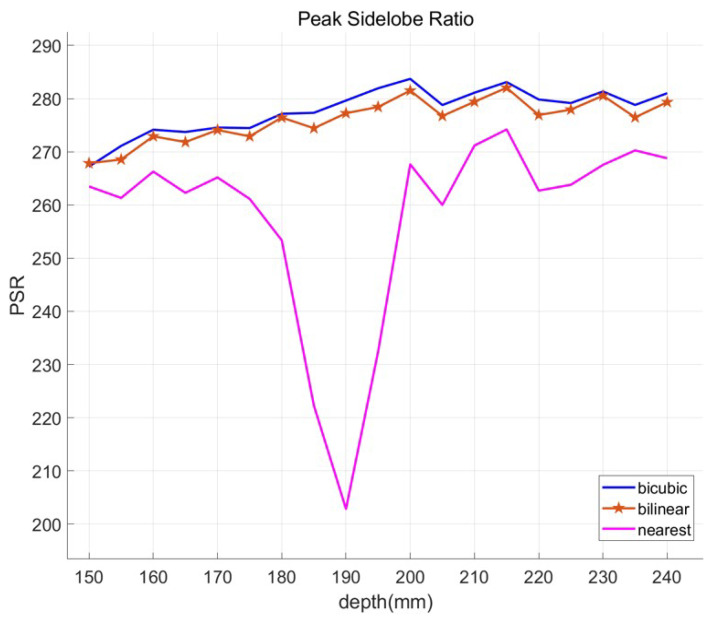
PSR results of simulation by digital zooming VCR with “bicubic” (blue line), “bilinear” (orange line), and “nearest” (magenta line) interpolation methods.

**Figure 11 sensors-23-02645-f011:**
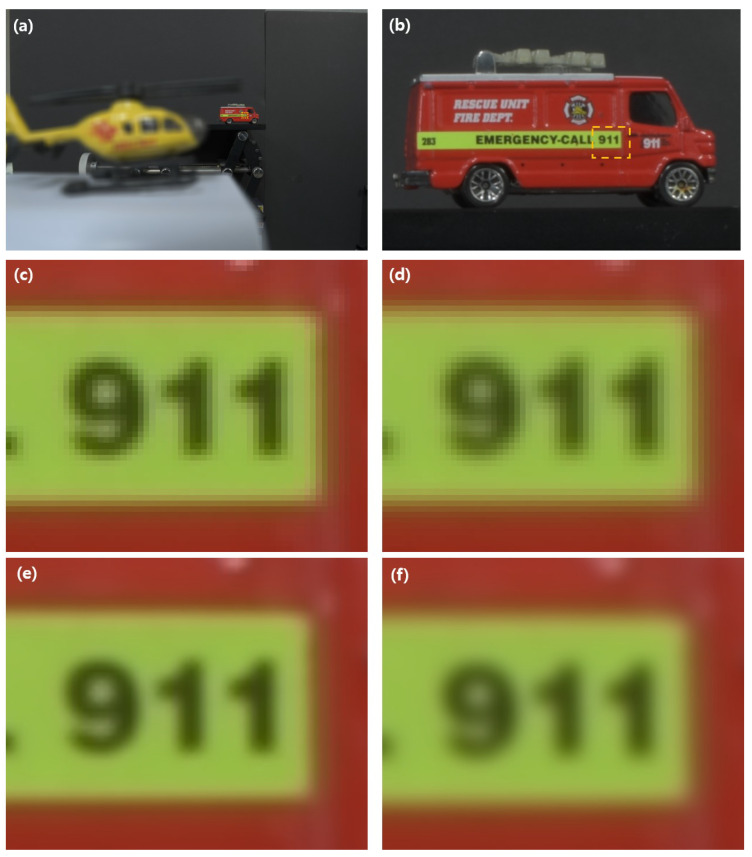
(**a**) Reconstructed three-dimensional image by conventional method, (**b**) reconstructed three-dimensional image by our proposed method, (**c**,**d**) cropped image from the results by conventional method at 344 mm and 392 mm, and (**e**,**f**) cropped image from the results by our proposed method at 43 mm and 49 mm, respectively.

**Figure 12 sensors-23-02645-f012:**
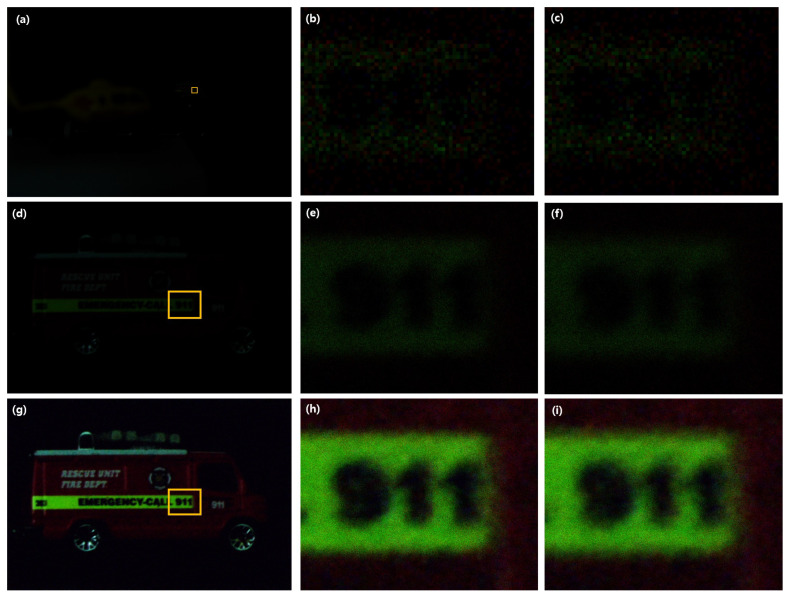
(**a**) Reconstructed three-dimensional image under photon-starved conditions by conventional method, (**b**,**c**) cropped images from results by computational photon counting imaging without digital zooming at 344 mm and 392 mm, (**d**) reconstructed three-dimensional image under photon-starved conditions by computational photon counting imaging with digital zooming, (**e**,**f**) cropped images from results by computational photon counting imaging with digital zooming at 43 mm and 49 mm, where 97,344 photons are used, (**g**) reconstructed three-dimensional image under photon-starved conditions by our method, where 97,344 photons and 10 observations are used, (**h**,**i**) cropped images from results by our method at 43 mm and 49 mm, respectively.

**Figure 13 sensors-23-02645-f013:**
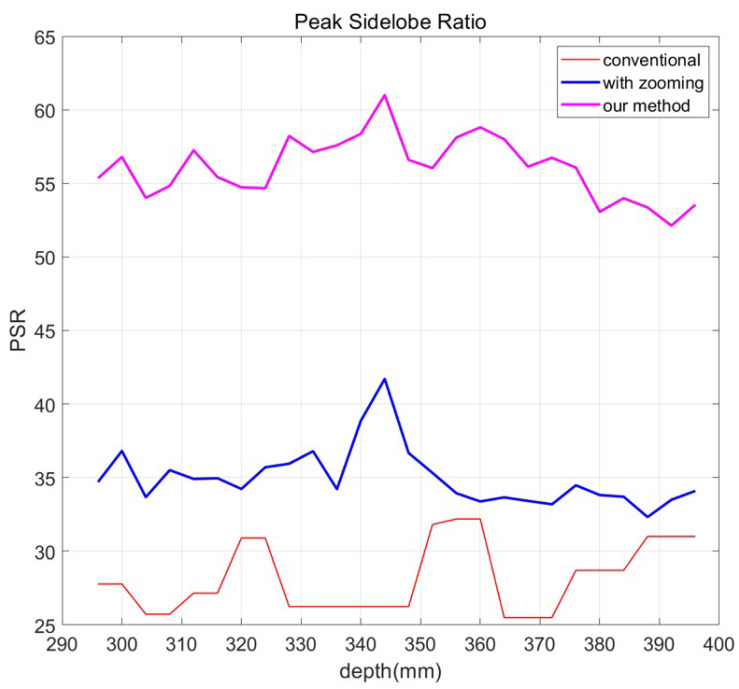
PSR results by computational photon counting integral imaging without digital zooming (conventional method, red line), computational photon counting integral imaging with digital zooming (blue line) and N observation photon counting integral imaging with digital zooming (magenta line).

**Table 1 sensors-23-02645-t001:** Processing time and PSR value for interpolation methods such as nearest, bilinear, and bicubic.

	Nearest	Bilinear	Bicubic
CPU Time (Photon-counting X)	1.538 s	1.539 s	1.563 s
PSR (Photon-counting X)	267.625	281.502	283.708

**Table 2 sensors-23-02645-t002:** PSR values of simulation results via various reconstruction depths by digital zooming with interpolation methods such as nearest, bilinear, and bicubic. Magnification ratio is 8.8.

Depth	Bicubic	Bilinear	Nearest
136.36	686.4	684.88	666.03
141.36	694.05	688.82	657.41
146.36	701.53	697.89	672.88
151.36	708.06	702.32	677.2
156.3	706.54	703.33	670.42
161.36	723.68	713.62	669.73
166.36	721.34	716.03	647.86
171.36	723.47	713.39	532.76
176.36	727.69	718.97	629.1
**181.36**	727.29	719.26	672.78
186.36	**727.47**	718.81	**705.19**
191.36	721.75	718.71	700.6
196.36	725.19	**720.57**	677.61
201.36	725.17	718.62	684.69
206.36	721.15	714.82	675.91
211.36	717.69	714.12	671.25
216.36	715.21	712.53	681.92
221.36	715.05	707.95	677.67
226.36	710.48	705.72	664.27

**Table 3 sensors-23-02645-t003:** Comparison of PSR values for interpolation methods, such as nearest, bilinear, and bicubic via reconstruction depths under photon-starved conditions.

Depth	Bicubic	Bilinear	Nearest
37 mm	60.369	**55.347**	56.645
37.5 mm	56.623	**56.79**	54.915
38 mm	57.572	**54.012**	55.894
38.5 mm	58.496	**54.822**	57.674
39 mm	59.836	**57.243**	57.977
39.5 mm	60.894	**55.426**	55.376
40 mm	55.991	**54.728**	55.693
40.5 mm	57.311	**54.661**	57.142
41 mm	58.025	**58.222**	60.222
41.5 mm	57.470	**57.132**	56.611
42 mm	57.428	**57.582**	53.567
42.5 mm	55.767	**58.363**	54.942
**43 mm**	57.045	**60.997**	54.663
43.5 mm	58.734	**56.593**	57.633
44 mm	58.733	**56.037**	56.383
44.5 mm	58.865	**58.117**	53.274
45 mm	57.778	**58.805**	53.966
45.5 mm	58.506	**58.003**	52.855
46 mm	58.043	**56.135**	52.302
46.5 mm	58.627	**56.737**	55.504
47 mm	59.309	**56.073**	55.979
47.5 mm	56.762	**53.066**	53.567
48 mm	57.923	**53.987**	52.069
48.5 mm	56.267	**53.361**	52.403
49 mm	57.754	**52.126**	54.06
49.5 mm	53.562	**53.546**	50.501

**Table 4 sensors-23-02645-t004:** Comparison among conventional imaging method, digital zooming method, and our proposed method via various reconstruction depths by PSR.

Depth (Conventional)	Conventional Method	With Digital Zooming	Our Method
37 mm (296 mm)	27.783	34.7	**55.347**
37.5 mm (300 mm)	27.783	36.812	**56.79**
38 mm (304 mm)	25.731	33.67	**54.012**
38.5 mm (308 mm)	25.731	35.507	**54.822**
39 mm (312 mm)	27.162	34.907	**57.243**
39.5 mm (316 mm)	27.162	34.956	**55.426**
40 mm (320 mm)	30.89	34.225	**54.728**
40.5 mm (324 mm)	30.89	35.7	**54.661**
41 mm (328 mm)	26.249	35.94	**58.222**
41.5 mm (332 mm)	26.249	36.793	**57.132**
42 mm (336 mm)	26.249	34.21	**57.582**
42.5 mm (340 mm)	26.249	38.861	**58.363**
43 mm (344 mm )	26.249	41.706	**60.997**
43.5 mm (348 mm)	26.249	36.669	**56.593**
44 mm (352 mm)	31.809	35.317	**56.037**
44.5 mm (356 mm)	32.183	33.931	**58.117**
45 mm (360 mm)	32.183	33.377	**58.805**
45.5 mm (364 mm)	25.502	33.653	**58.003**
46 mm (368 mm)	25.502	33.412	**56.135**
46.5 mm (372 mm)	25.502	33.185	**56.737**
47 mm (376 mm)	28.709	34.474	**56.073**
47.5 mm (380 mm)	28.709	33.81	**56.066**
48 mm (384 mm)	28.709	33.698	**53.987**
48.5 mm (388 mm)	30.993	32.317	**53.361**
49 mm (392 mm)	30.993	33.493	**52.126**
49.5 mm (396 mm)	30.993	34.092	**53.546**

## Data Availability

Not applicable.

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
