# Peer review of "Three-Dimensional Digital Zooming of Integral Imaging under Photon-Starved Conditions"

_sensors, 2023, doi:10.3390/s23052645_

Round 1

Reviewer 1 Report

This manuscript presents the digital zooming method of integral imaging under photon-starved conditions. I think it is well-written and well-organized. However, I have several concerns as the follows:

1.     Which image is used as reference for calculating PSR?

2.     Why is the exposure time 13 seconds?

3.     When interpolation is used in zooming method, which interpolation is used?

4.     Is there any reason why the yellow helicopter toy is located at 40mm?

5.     Why is the pitch 0.5mm?

If authors can follow my comments, I can recommend this manuscript as publication.

Reviewer 2 Report

No comments.

Reviewer 3 Report

The paper proposes a 3D digital zooming to improve image visualization at long distance under Photon-Starved Conditions.

The paper has numerous shortcomings and needs to be revised. Moreover, its novelty is rather limited since the only innovative elements are simple formulas (13-20).

The main problems to face are as follows.

The problem connected to photon-starved conditions is not well framed in the introduction. These conditions are mentioned without adequately having introduced them before. Consequently, it is difficult for a reader, not an expert, to understand the text. This should be better introduced and described.

Always in the Introduction, the authors should insert a preliminary discussion of what major drawbacks of existing methods they intend to address by highlighting the novelty of the method and the gap that is being closed. This part should be rewritten and better explained by reporting and extending the considerations at lines 36-80, Page 2.

In Section 2.1, the authors need to mention the references to the interpolation method they use and the Deep Learning based methods they refer to (see lines 76-83 on page 2). In addition, this related work section should be better explained and deepened. In particular, the existing methods with similar characteristics to the one proposed should be summarily described (and then used in the validation phase). Since the proposed method is based on an interpolation scheme, I suggest considering some existing methods belonging to this category (for example, recently proposed methods based on Lagrange and de la Vallée-Poussin interpolation scheme).

Also, Section 2.2 should be carefully revised and rewritten in light of the above considerations made for Section 2.1, although the references are present here.

Section 2.3 and 2.4.3should probably be a subsection of Section 3, and Section 2 should be differently titled, for example, as “Related work”.

The proposed method's description also needs to be improved as many aspects are omitted, ambiguous, and missing. In particular, the novelty of the proposed method should be highlighted in comparison to other similar existing methods. In this Section, also the main features of the proposed method should be better indicated. For instance, if not integer scale factors can be used, the complexity of the algorithm, and so on, should be reported.

The most critical part of the paper concerns the experimental validation of the proposed method since it results to be completely unconvincing. The design of the validation process needs to be greatly improved. Since it is impossible here to indicate all aspects that should be considered (it is probably better to draw inspiration from more recent papers), only a limited selection is proposed in the following.

-  - The number and type of datasets should necessarily be expanded so that the method validation can be considered significant.

- - Numerous benchmark methods should be considered, and a deep comparison with other methods should be made. The figure caption of Figure 8 cites “conventional methods,” but nothing else is given in the whole paper.

-  -  The experimentation should be done for many different parameters, and the results should be evaluated at the parameters.

-   - The employed Metric PSR should be defined, and the quantitative results should be better reported for all datasets. Besides Figure 10, some tables containing experimental data should be added.

- - Visual comparison alone on an image of a given class does not help determine the quality of the method's performance and compare it with other benchmark methods. I suggest improving significantly this part devoted to the qualitative performance evaluation. Moreover, Figures 4, 7, and 9 are not adequately visible, while Figure 3 is too big, and the results in Figure 8 are not visually good.

-    -  The CPU time and computational complexity are aspects that should be considered during the comparison with the other methods. These should be reported in any shown experimental result

-   -   All experimental results should also be shown for high-scale factors.

--  The limitations of the proposed method, as well as the future improvements, should be extensively discussed.

Round 2

Reviewer 3 Report

The changes/additions made by the authors are insufficient and inadequate to address the reported critical points.

The problem connected to photon-starved conditions remain poorly framed since only a few comments have been added. Moreover, in the introduction, the authors do not insert a preliminary discussion of what major drawbacks of existing methods they intend to address by highlighting the novelty of the method and the gap that is being closed.

In Section 2.1, the authors do not mention the references to the interpolation method they use and the Deep Learning based methods they refer to (see lines 76-83 on page 2). Moreover, this related work section needs to be better explained and deepened. In particular, the existing methods with similar characteristics to the one proposed are not summarily described (and then used in the validation phase).

Regarding the interpolation scheme, according to the revised text, the authors demonstrate inadequate knowledge of the literature in the field. Contrary to what the authors claim, among those the classical interpolation schemes, bicubic interpolation is adopted by almost all authors as a benchmark method because it has the best performance and is the fastest. However, this method has some drawbacks, and new recent methods have been introduced that outperform it. Please, for instance, consider the following methods whose code is publicly available:

Deep networks for image super-resolution with sparse prior (2015)

Lagrange–Chebyshev Interpolation for image resizing (2022)

Image scaling by de la Vallée-Poussin filtered interpolation (2022)

The proposed method's description has not been improved, as suggested.

The experimental validation of the proposed method results improved at the formal level, but, at the methodological level, it is not satisfactory because many reported issues have not been addressed and the benchmark methods are too dated.

There are some spelling and punctuation errors. In particular, sentences starting with “. And” should be eliminated.
